# Machine Learning-Based Predictive Model of Aortic Valve Replacement Modality Selection in Severe Aortic Stenosis Patients

**DOI:** 10.3390/medsci12010003

**Published:** 2023-12-29

**Authors:** Ronpichai Chokesuwattanaskul, Aisawan Petchlorlian, Piyoros Lertsanguansinchai, Paramaporn Suttirut, Narut Prasitlumkum, Suphot Srimahachota, Wacin Buddhari

**Affiliations:** 1Division of Cardiovascular Medicine, Department of Medicine, Faculty of Medicine, Center of Excellence in Arrhythmia Research, Chulalongkorn University, Bangkok 10330, Thailand; drronpichaic@gmail.com (R.C.); taepiyoros@gmail.com (P.L.); chertam-87@hotmail.com (P.S.); s_srimahachota@yahoo.co.th (S.S.); wacin_buddhari@yahoo.com (W.B.); 2Cardiac Center, King Chulalongkorn Memorial Hospital, The Thai Red Cross Society, Bangkok 10330, Thailand; 3Division of Geriatric Medicine, Department of Medicine, Faculty of Medicine, Chulalongkorn University, Bangkok 10330, Thailand; aisawan.p@chula.ac.th; 4Geriatric Excellence Center, King Chulalongkorn Memorial Hospital, The Thai Red Cross Society, Bangkok 10330, Thailand; 5Department of Cardiovascular Medicine, Mayo Clinic, Rochester, MN 559020, USA

**Keywords:** aortic valve replacement, decision tree, LASSO, SAVR, TAVR

## Abstract

The current recommendation for bioprosthetic valve replacement in severe aortic stenosis (AS) is either surgical aortic valve replacement (SAVR) or transcatheter aortic valve replacement (TAVR). We evaluated the performance of a machine learning-based predictive model using existing periprocedural variables for valve replacement modality selection. We analyzed 415 patients in a retrospective longitudinal cohort of adult patients undergoing aortic valve replacement for aortic stenosis. A total of 72 clinical variables including demographic data, patient comorbidities, and preoperative investigation characteristics were collected on each patient. We fit models using LASSO (least absolute shrinkage and selection operator) and decision tree techniques. The accuracy of the prediction on confusion matrix was used to assess model performance. The most predictive independent variable for valve selection by LASSO regression was frailty score. Variables that predict SAVR consisted of low frailty score (value at or below 2) and complex coronary artery diseases (DVD/TVD). Variables that predicted TAVR consisted of high frailty score (at or greater than 6), history of coronary artery bypass surgery (CABG), calcified aorta, and chronic kidney disease (CKD). The LASSO-generated predictive model achieved 98% accuracy on valve replacement modality selection from testing data. The decision tree model consisted of fewer important parameters, namely frailty score, CKD, STS score, age, and history of PCI. The most predictive factor for valve replacement selection was frailty score. The predictive models using different statistical learning methods achieved an excellent concordance predictive accuracy rate of between 93% and 98%.

## 1. Introduction

Aortic stenosis (AS) is a major cardiovascular problem which is projected to affect a significant portion of the older population [1]. Surgical aortic valve replacement (SAVR) has traditionally been considered the standard treatment. Over the past 10 years, transcatheter aortic valve replacement (TAVR) has emerged as a non-surgical alternative modality that can provide comparable efficacy and safety for selected severe AS patients [2]. The current recommendation for severe AS patients who have been approved to receive a bioprosthetic valve is either SAVR or TAVR. Patients with prohibitive risk for SAVR are recommended for either TAVR or palliative care as a destination therapy [3].

SVAR’s efficacy is supported by more than 50 years of experience, with sufficient data across different age groups. While TAVR is a non-surgical option that brings benefits including a lower risk of all-cause death and atrial fibrillation, valve durability continues to be a major concern. Balancing the goals of patient longevity and valve durability is critical in making a valve replacement modality selection [4].

Several patient-specific factors also have been incorporated for the final decision based on a shared decision-making process by a multidisciplinary heart team consisting of at least a non-invasive cardiologist, an interventional cardiologist, and a cardiac surgeon [5,6]. A definite consensus on the first-line therapy has not yet been achieved for the patient age group between 65 and 85 years [3]. Increased use of predictive models has become a significant part of health management decisions [7,8,9,10]. Currently, data on factors related to aortic valve replacement remain limited.

Our study aims to develop predictive models to assess readily available patient information collected during the preprocedural assessment, which includes demographic, clinical, laboratory investigation, and cardiac imaging data that impact the selection of an aortic valve replacement modality. Using medical records from a university hospital in Thailand, these models reflect the decision-making process for patients older than 65 years in a high-volume tertiary center in a developing country.

## 2. Materials and Methods

We conducted a retrospective chart-review study of older patients with symptomatic severe aortic stenosis who consecutively underwent TAVR or SAVR between January 2010 and December 2020, based on our own institution databases as stated in prior publications [11]. The study was carefully appraised and approved by the Ethics Review Committee. Inclusion criteria were older adults aged 65 years old and above. Variables on demographic and preoperative clinical characteristics were extracted from electronic medical records, including comorbidities, cardiovascular intervention history, medications, symptoms of aortic stenosis, baseline 12-lead electrocardiography (ECG), baseline echocardiographic characteristics, the Society of Thoracic Surgery (STS) risk score, and frailty score [12].

### 2.1. Treatment of Missing Data

To ascertain the randomness of missing data, analysis of missing data distribution was performed for quality assurance. Variables with more than 5% missing values were excluded. For the remaining variables, random forest method was employed to impute on any missing data, for which repeated procedures were executed on each iteration.

### 2.2. Model Selection and Creation

In each round of iteration, 20% of subjects with complete data were randomly separated to serve as a test set. The remaining 80% along with imputed data formed the training set. Following the training set with a 10-fold cross-validation method, the model predicting intervention modality could be generated. The resulting model was tested for precision using the test set. These processes were repeated for 100 iterations (Figure 1).

Machine learning models used in this study were LASSO regression models and decision tree models. To ensure the robustness and validity of individual essential features, the best models generated from each iteration were archived and compared. Models derived from LASSO regression were displayed with coefficients. For decision trees, the percentage of deviation improvement of each variable was displayed. Concordance index (C-index) was utilized for precision measurement on both models.. Data were analyzed with R version 4.0.5 using packages “randomForest”, “glmnet”, and “rpart” [11].

### 2.3. Outcomes

The outcome of interest in this study was the precision of the model on the prediction of aortic valve replacement modality selected for the patient. The two possible aortic valve replacement modalities included either a surgical aortic valve replacement (SAVR) or a transcatheter aortic valve replacement (TAVR).

## 3. Results

### 3.1. Study Population

From January 2010 to December 2020, 415 participants were included, with 238 (57%) receiving SAVR and 177 (43%) receiving TAVR. Table 1 shows the preoperative characteristics of the SAVR versus TAVR group. TAVR patients were older, frailer, had a higher STS score, and had worse heart failure status compared to those in the SAVR group.

### 3.2. Model Derivation

We conducted a LASSO regularization and decision tree analysis to develop models to predict intervention modality selection. As shown in Table 2 and Table 3, we demonstrated the variables with their respective discrimination ability, ordering them by their frequency of being included in the models. Age, STS score, frailty score, and CKD were the most frequent variables included in models derived from both methods.

#### Model Validation

We used C-index to evaluate the precision of the prediction models (LASSO in Table 2, decision tree in Table 3 and Figure 1) on the test set. The results of our models were excellent, with 98.6% and 93% precision for LASSO and decision tree, respectively.

The precision of the LASSO model and the coefficients of the 10 most frequent variables derived from 100 iterations are displayed in Table 2. The most precise model yielded 98% accuracy. The median precision of all models was 92% (IQR 89–93%). The variables present in all models were age, STS score, frailty score, and CKD.

Next, the precision of the decision tree model and the improvement in the deviation of the 10 most frequent variables derived from 100 iterations are illustrated in Table 3. The best decision tree model yielded 93% precision. The median precision of all models was 86% (IQR 83–88%). Variables presented in all models included age, STS score, frailty score, and CKD.

Figure 2 shows the best decision tree model as an example of how decision trees predict intervention modality. High frailty score, high STS score, and presence of CKD aided in the decision to choose TAVR.

## 4. Discussion

In our study, we used observational data to construct a model predicting the selection of aortic valve replacement modality (SAR vs. TAVR) in our institution in Thailand. The results represent one of the first predictive models on this aspect among developing countries. Our model relies on fundamental information routinely obtained during preprocedural assessment in tertiary care settings among patients eligible for bioprosthetic aortic valve replacement. Generalizability of model application is a key strength of our model. This model could aid in the shared decision making between care providers and patients in selecting a suitable valve replacement modality. Furthermore, in non-conventional scenarios, in which patients’ characteristics are non-parallel to clinical trials, our model could be beneficial in providing a greater confidence level to modality selection guidance.

Current recommendations on valve selection between TAVR and SVAR are based primarily on the patient’s clinical profile. In symptomatic or asymptomatic left ventricular systolic dysfunction (LVEF < 50%) with severe AS, favorable factors for choosing SAVR are younger patients (age < 65 years) with longevity greater than 20 years and challenging vascular anatomy for TAVR [13]. In the absence of anatomical considerations for TAVR, elderly patients who are older than 80 years or younger patients with a life expectancy of less than 10 years are preferable candidates for TAVR. Shared decision making should be made for patients over 65 years old since both TAVR and SAVR show merit as a first line recommendation. Valve selection for this age category continues to be very challenging and assessing data to determine factors related to selection modality can help inform both doctors and patients. Our study demonstrates the great value of information on the local clinical practice, representing limited resources, countries, and local patient population.

Owing to the growing demand for transcatheter aortic valve replacement (TAVR), numerous less-experienced medical centers are now undertaking this procedure. However, most validated models are derived from highly proficient centers, potentially lacking a representation of real-world evidence in settings with limited resources. Despite the acknowledged increased surgical risk associated with additional factors such as CKD, these elements are not explicitly integrated into decision-making algorithms for selecting between TAVR and surgical aortic valve replacement (SAVR). Clinical decisions often rely on the expertise of the operator, which may not be in line with established scientific guidelines. This study illustrates that predictors observed in settings with lower case volumes align with recommendations in guidelines, establishing a quality control metric for real-world clinical practices in resource-constrained environments.

The utilization of artificial intelligence (AI) in the preprocedural planning of TAVR through CT scans has been demonstrated to diminish interobserver reliability [14]. The outcomes of this investigation contribute significantly to enhancing preoperative assessments, particularly in less-experienced medical centers. Consequently, this fosters an equitable distribution of TAVR procedures across centers with varying levels of expertise. The primary objective of this approach is to alleviate healthcare disparities on a global scale, mirroring our own study’s intent to elucidate the factors influencing the determination of a patient’s eligibility for aortic valve replacement.

Moreover, the assessment of preprocedural extra-aortic valve abnormality parameters and their subsequent measurements plays a crucial role in determining the long-term prognosis [15]. Notably, irreversible right ventricular (RV) dysfunction emerges as a defining factor delineating patients with a poorer long-term prognosis when juxtaposed with those devoid of RV dysfunction. It is imperative to underscore that these intricate findings may elude detection through the application of AI, given that such factors may initially appear inconspicuous within the scope of the available knowledge at that juncture.

The utilization of artificial Intelligence (AI) in the processing of preprocedural multimodality imaging, as elucidated in a study by Maier et al. [16], stands as a noteworthy initiative aimed at enhancing patient outcomes. Notably, our approach aligns with this paradigm, specifically in the context of discerning the suitability of patients for transcatheter aortic valve replacement (TAVR) or surgical aortic valve replacement (SAVR) within a clinical framework analogous to our own. The model, distinguished by its capacity to be wielded even by operators with limited experience, emerges as a valuable tool for selecting patients eligible for TAVR procedures. This not only streamlines the decision-making process but also contributes to a reduction in procedural steps and the overall healthcare burden, thereby maximizing the utility of readily available information.

Our cohort’s uniqueness stems from the selection of patients with very high cardiovascular risks and compares them to other high-risk Asian cohorts with a comparable 1-year survival rate of 84.7% and 88.6% for TAVR and SAVR, respectively [17,18].

From the variable selection results, our study showed that age, STS score, frailty score, and CKD are the highest-value predictors. Our findings emphasize distinct predictors of valve selection in which accessible data collected at the point of service were utilized to generate the prediction model.

The decision process for valve selection has normally depended on the heart valve team in collaboration with the patient and their family [19]. Our study identifies factors associated with these modality decisions. Frailty, STS, and CKD are important variables selected by the model that achieved 94% accuracy in the prediction of valve modality. These factors reflect factors that clinicians weigh in real-world practice and further explore in detail based on what the standard guideline has already recommended [3]. In addition to our model’s performance in deriving related factors, our cohort demonstrated similar outcome profiles compared to other clinical cohorts, which indeed supported the efficiency of the decision model to maintain the status quo. These factors should be prospectively studied to further explore their significance on the outcome.

Whether these significant parameters are justified to be a part of the patient selection criteria remains unknown and will need a long-term study to explore the relationship between the predictive factors and long-term outcomes. Due to the current unavailability of outcome data, it is important to begin compiling clinical and demographic characteristics into a large registry database containing future procedure and postprocedural care data as well as outcome data.

Despite expert-opinion-led decision making on valve replacement for individual patients being considered as a standard practice, the rationale for each decision under different circumstances has never been documented. This model pinpoints the clinically important factors that experts used to determine a suitable valve replacement modality. Therefore, those factors may be incorporated into a local guideline and used in assisting the design of further studies to explore the impact on long-term outcomes.

Our study has limitations that must be acknowledged. Firstly, the enrolled subjects from a single center might not represent the patient characteristics in developed countries. Furthermore, the present study had a small sample size, which is not feasible for all machine learning methods. However, the methods used in this study (i.e., imputation, LASSO regression, decision tree) are valid with a small sample size. Also, our center is one of Thailand’s highest TAVR volume centers.

## 5. Conclusions

In conclusion, we have described our machine learning models predictions of the probability of aortic valve replacement modality selection among our long-term cohort. This study is among the very first to present models that reflect in-depth clinical practice in a developing country. We also constructed and validated the tool to predict valve replacement modality selection. These local data can help address a range of health care-associated factors that impact decision making in the valve selection process. Information from this model might also be used to determine patient care quality and shape future study designs that can optimize patient selection criteria. The results could serve as an assistance tool to guide clinicians in their assessment of important clinical variables that impact the selection of the optimal valve replacement procedure based on an individual’s clinical profile in resource-constrained clinical settings. Further study is required to explore the clinical benefits of the predictive model and factors in clinical practice.

## Figures and Tables

**Figure 1 medsci-12-00003-f001:**
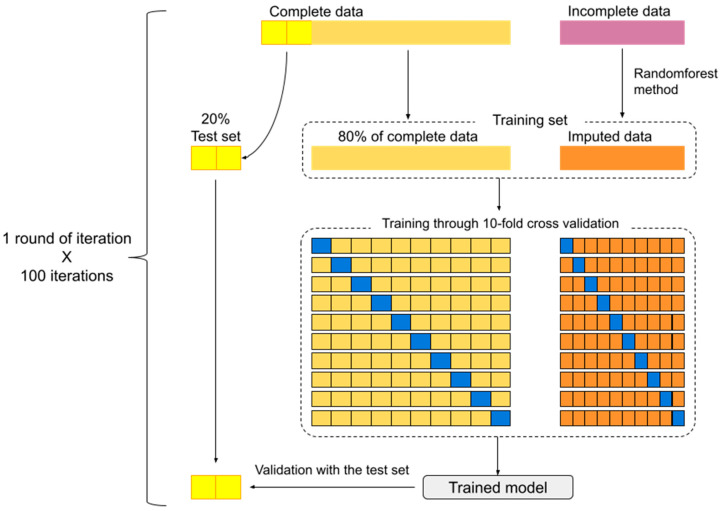
Model development and 100 iterations scheme. (The excluded data (10% of dataset) during bootstrap method is symbolized by the blue block).

**Figure 2 medsci-12-00003-f002:**
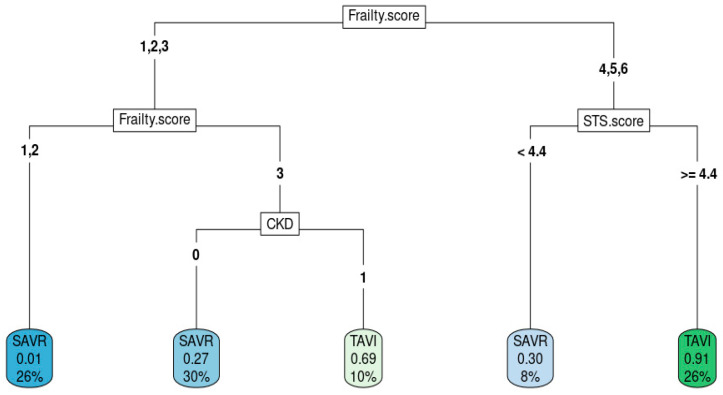
Decision tree from the best model.

**Table 1 medsci-12-00003-t001:** Baseline characteristics of patients in SAVR and TAVR cohorts.

	Overall	SAVR	TAVI	*p*
*n*	415	238	177	
Age (mean (SD))	78.41 (7.41)	75.41 (6.50)	82.44 (6.63)	<0.001
Gender (%)	220 (53.0)	121 (50.8)	99 (55.9)	0.353
STS.score (mean (SD))	5.63 (4.27)	3.97 (1.73)	7.89 (5.49)	<0.001
Frailty.score (mean (SD))	3.26 (1.06)	2.71 (0.78)	4.01 (0.93)	<0.001
Frailty score > 4 = TRUE (%)	150 (36.1)	31 (13.0)	119 (67.2)	<0.001
NYHA (%)				<0.001
0	2 (0.5)	0 (0.0)	2 (1.1)	
1	4 (1.0)	4 (1.7)	0 (0.0)	
2	196 (47.2)	134 (56.3)	62 (35.0)	
3	192 (46.3)	94 (39.5)	98 (55.4)	
4	20 (4.8)	6 (2.5)	14 (7.9)	
CAD (%)	113 (27.2)	55 (23.1)	58 (32.8)	0.038
AF AFL (%)	57 (13.7)	19 (8.0)	38 (21.5)	<0.001
CKD (%)	141 (34.0)	27 (11.3)	114 (64.4)	<0.001
HF (%)	234 (56.4)	91 (38.2)	143 (80.8)	<0.001
COPD (%)	33 (8.0)	10 (4.2)	23 (13.0)	0.002
Smoke (%)	19 (4.6)	7 (2.9)	12 (6.8)	0.107
HT (%)	339 (81.7)	191 (80.3)	148 (83.6)	0.455
DM (%)	144 (34.7)	73 (30.7)	71 (40.1)	0.058
DLP (%)	297 (71.6)	162 (68.1)	135 (76.3)	0.085
LVEF (mean (SD))	62.52 (15.83)	63.18 (15.53)	61.67 (16.23)	0.343
TVD (%)	89 (21.4)	45 (18.9)	44 (24.9)	0.168
Calcify.Ao (%)	22 (5.3)	1 (0.4)	21 (11.9)	<0.001
CABG (%)	38 (9.2)	2 (0.8)	36 (20.3)	<0.001
RVSP (mean (SD))	22.22 (21.62)	18.57 (21.35)	26.83 (21.14)	<0.001
mPAP (mean (SD))	1.52 (6.40)	0.82 (4.73)	2.19 (7.63)	0.057
MS…Mod (%)	7 (1.7)	1 (0.4)	6 (3.4)	0.007
MR…Mod (%)	35 (8.4)	12 (5.0)	23 (13.0)	0.003
MR…Severe (%)	4 (1.0)	1 (0.4)	3 (1.7)	0.05

Abbreviation: AF, atrial fibrillation; AFL, atrial flutter; Ao, aorta; CABG, coronary artery bypass graft; CAD, coronary artery disease; CKD, chronic kidney disease; COPD, chronic obstructive pulmonary disease; DLP, dyslipidemia; DM, diabetes mellitus; HF, heart failure; HT, hypertension; LVEF, left ventricular ejection fraction; mPAP, mean pulmonary arterial pressure; MR, mitral regurgitation; MS, mitral stenosis; NYHA, New York Heart Association; RVSP, right ventricular systolic pressure; TVD, triple vessel disease.

**Table 2 medsci-12-00003-t002:** Parameter coefficients obtained through LASSO regression.

	Best Model	Median	1st Quartile	3rd Quartile	Frequency
Precision	98.6%	92.1%	89.5%	93.4%	-
Age	−0.0703	−0.0903	−0.1286	−0.0697	100
STS.score	−0.2019	−0.2688	−0.3731	−0.1797	100
Frailty.score2	1.7302	2.2415	1.9649	2.6769	100
Frailty.score4 [MU4]	−0.5575	−0.9194	−1.2795	−0.6159	100
CKD1	−1.0407	−1.7186	−2.124	−1.481	100

**Table 3 medsci-12-00003-t003:** Best model parameters derived from the decision tree method.

	Best Model	Median	1st Quartile	3rd Quartile	Frequency
Precision	93%	86%	83%	88%	-
Age	0.08	0.09	0.08	0.11	100
CKD	0.11	0.15	0.13	0.32	100
Frailty score	0.39	0.29	0.2	0.33	100
STS.score	0.23	0.2	0.18	0.22	100
CABG	0.03	0.04	0.02	0.05	86
NYHA	0.02	0.01	0	0.01	82

## Data Availability

Data sharing will be available upon request to the corresponding author.

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
