# Peer review of "Machine Learning-Based Predictive Model of Aortic Valve Replacement Modality Selection in Severe Aortic Stenosis Patients"

_medsci, 2023, doi:10.3390/medsci12010003_

Round 1

Reviewer 1 Report

Comments and Suggestions for Authors

This is a well written manuscript. You can find my suggestions below

Major comments

1) What is the added value of your findings, since the AI algorithm revealed the predictors that we already knew to assist in decision making regarding SAVR/TAVR selection

2) Discussion is too short and the same holds true for the references included in this section. You are kindly requested to go through literature again and to include some relevant studies in the field (e.g doi: 10.1016/j.ebiom.2023.104794/ 10.1136/openhrt-2022-002068/10.1155/2022/1368878. eCollection 2022/ 10.1016/j.jcin.2020.09.011.)

Minor Comments

1) Figure 1: Please check randomforest is it correct or should you make it random forest?

2) Table 1: Are there data regarding EuroScore?

3) Table 1: provide the definitions of the utilized abbreviations in Table 1

4) Which score was used to define frailty (please provide a detailed description of the scoring system or provide an appropriate reference instead)

Comments on the Quality of English Language

Use of English is fine

Author Response

We appreciate the valuable comments from the reviewer. Our responses are as below

Major comments

  • What is the added value of your findings, since the AI algorithm revealed the predictors that we already knew to assist in decision making regarding SAVR/TAVR selection

Response:

Thank you for the reviewer’s response. Due to the increasing demand for Transcatheter Aortic Valve Replacement (TAVR), numerous low-volume centers are now performing this procedure. However, most validated models stem from highly experienced centers, potentially lacking a reflection of real-world evidence in resource-constrained settings. Despite the acknowledged heightened surgical risk associated with additional predictors like Chronic Kidney Disease (CKD), these factors are not explicitly incorporated into decision-making algorithms for choosing between TAVR and Surgical Aortic Valve Replacement (SAVR). Clinical decisions often hinge on operator experience, which may not align with established scientific guidelines. This study demonstrates that predictors observed in lower-volume settings align with guideline recommendations, establishing a quality control measure for real-world clinical practices in resource-limited environments. The following text has been inserted into the discussion section.

Owing to the growing demand for Transcatheter Aortic Valve Replacement (TAVR), numerous less-experienced medical centers are now undertaking this procedure. However, most validated models are derived from highly proficient centers, potentially lacking a representation of real-world evidence in settings with limited resources. Despite the acknowledged increased surgical risk associated with additional factors such as Chronic Kidney Disease (CKD), these elements are not explicitly integrated into decision-making algorithms for selecting between TAVR and Surgical Aortic Valve Replacement (SAVR). Clinical decisions often rely on the expertise of the operator, which may not be in line with established scientific guidelines. This study illustrates that predictors observed in settings with lower case volumes align with recommendations in guidelines, establishing a quality control metric for real-world clinical practices in resource-constrained environments.

  • Discussion is too short and the same holds true for the references included in this section. You are kindly requested to go through literature again and to include some relevant studies in the field (e.g doi: 10.1016/j.ebiom.2023.104794/ 10.1136/openhrt-2022-002068/10.1155/2022/1368878.eCollection 2022/ 10.1016/j.jcin.2020.09.011.)

Response: We appreciate your comment. Following your input, we conducted additional literature research and integrated the latest evidence relevant to our study into the discussion section.

Minor Comments

  • Figure 1: Please check randomforest is it correct or should you make it random forest?

Response: Thank you for the comment. This term in the caption for Figure 1 is already correct. (https://cran.r-project.org/web/packages/randomForest/)

  • Table 1: Are there data regarding EuroScore?

Response: We appreciate the reviewer's comment. In accordance with the guidelines from the American College of Cardiology/American Heart Association (ACC/AHA) and the European Society of Cardiology/European Association for Cardio-Thoracic Surgery (ESC/EACTS) regarding the management of valvular heart disease, a critical element in determining the appropriate treatment for severe symptomatic aortic valve stenosis using TAVR entails assessing the inherent risk linked to SAVR. This assessment is partly influenced by factors such as the Society of Thoracic Surgeons (STS) score or the European System for Cardiac Operative Risk Evaluation (EuroSCORE) II. By employing the STS score, patients can be grouped into three risk categories. To tackle concerns related to multicollinearity in the model development process, we integrated the STS into the model due to its convenience and comparable predictive capabilities to EuroSCORE.

Reference:

Nishimura RA, Otto CM, Bonow RO, et al. 2017 AHA/ACC focused update of the 2014 AHA/ACC guideline for the management of patients with valvular heart disease: a report of the American College of Cardiology/American Heart Association Task Force on Clinical Practice Guidelines. Circulation. 2017;135:e1159-e1195.

  • Table 1: provide the definitions of the utilized abbreviations in Table 1

Response: Thank you for the observation. We have inserted the abbreviation in the table caption.

  • Which score was used to define frailty (please provide a detailed description of the scoring system or provide an appropriate reference instead)

Response: We appreciate this point. We have added a reference to the materials and methods section.

Reviewer 2 Report

Comments and Suggestions for Authors

The manuscript very clearly presents a machine learning model based on retrospective data for selection of SAVR over TAVR in a patient cohort >65 y/o.  This paper showcases the way in which machine learning can assist in clinical decision-making thus making this translational research relevant to readers of Medical Sciences.  The authors have identified several limitations to the study, but this does not take away from the technique delineating personalized treatment pathways wherein others who may adopt/adapt this can expand and empower healthcare practitioners with a tool to navigate the complexities of patient care.  The manuscript entertains the evaluation of diverse set parameters, reducing significant variable to a manageable set for aortic valve cardiac conditions, offers a tailored approach for individualized patient care.

 It would have been informative to know of other machine learning algorithms the researchers had explored, as this would also inform the readership.

The authors suggest the data is available on request, and it is curious why this might not be possible through, say, GitHub,

Author Response

We appreciate the valuable comments from the reviewer. Below are our responses

The manuscript very clearly presents a machine learning model based on retrospective data for selection of SAVR over TAVR in a patient cohort >65 y/o.  This paper showcases the way in which machine learning can assist in clinical decision-making thus making this translational research relevant to readers of Medical Sciences.  The authors have identified several limitations to the study, but this does not take away from the technique delineating personalized treatment pathways wherein others who may adopt/adapt this can expand and empower healthcare practitioners with a tool to navigate the complexities of patient care.  The manuscript entertains the evaluation of diverse set parameters, reducing significant variable to a manageable set for aortic valve cardiac conditions, offers a tailored approach for individualized patient care.

 It would have been informative to know of other machine learning algorithms the researchers had explored, as this would also inform the readership.

The authors suggest the data is available on request, and it is curious why this might not be possible through, say, GitHub,

Response: We appreciate the comment, In our manuscript, we employ the LASSO and decision tree algorithms because they are widely understood by non-statistician readers and commonly utilized in clinical pathway decision-making. The LASSO regression offers insights into the importance of variables, while the decision tree model involves examining the top 10 best c-statistics models and selecting the one most clinically relevant.

Reviewer 3 Report

Comments and Suggestions for Authors

Chokesuwattanaskul et colleagues evaluated the performance on machine learning to predict valve replacement in both TAVR and SAVR. The paper is well written and the investigation reguarding the prediction trough machine learning is an important topic that warrants further investigation and is a great clinical question. I support the acceptance for publication of the manuscript.

Author Response

Response: Thank you so much for your kind response. We believe that our manuscript will provide valuable information for the readers.

Round 2

Reviewer 1 Report

Comments and Suggestions for Authors

No further comments, congratulations for the overall efforts